# Human Placenta MSC-Derived DNA Fragments Exert Therapeutic Effects in a Skin Wound Model via the A2A Receptor

**DOI:** 10.3390/ijms26041769

**Published:** 2025-02-19

**Authors:** Hankyu Lee, Hyun-Jung Lee, Hyeon-Jun Jang, Hyeri Park, Gi Jin Kim

**Affiliations:** 1Department of Bioinspired Science, CHA University, Seongnam-si 13488, Republic of Korea; hglee@plabiologics.com (H.L.);; 2PLABiologics Co., Ltd., Seongnam-si 13522, Republic of Korea; hjlee@plabiologics.com (H.-J.L.); hjjang@plabiologics.com (H.-J.J.)

**Keywords:** mesenchymal stem cells (MSCs), DNA fragment, UNIPlax, wound healing, cell migration, angiogenesis

## Abstract

PDRN, polydeoxyribonucleotide, which is used as a tissue-regeneration material, is present in human cells under physiological conditions and stimulates regeneration and metabolic activity. PDRN can be used as a biomaterial for several types of regeneration, including wound healing, to promote cell growth and growth-factor production. The aims of this study were to determine the effect of PDRN derived from human placenta-derived mesenchymal stem cells (hPD-MSCs) on cellular regeneration through A2A receptor signaling and to investigate its therapeutic effects in a mouse model of wound healing. Human PDRN (UNIPlax) was extracted from hPD-MSCs fragmented via a sonication system and evaluated for its effect on the migration of HaCaT cells in an in vitro system and in a wound-healing mouse model in vivo. Compared with the sham treatment, UNIPlax treatment significantly increased the migration of injured HaCaT cells (*p* < 0.05). Additionally, the tube formation of human umbilical vein endothelial cells (HUVECs) was greater than that of the sham group (*p* < 0.05), and the effects of this treatment were mediated through the A2A receptor. Furthermore, UNIPlax treatment led to a decrease in wound size; in addition, the area of granulation and the rate of collagen formation at the wound site were significantly greater than those in the sham group in the wound-healing mouse model (*p* < 0.001). We also confirmed that UNIPlax promoted tissue regeneration and the expression of VEGF through the A2A receptor. Taken together, these findings indicate that UNIPlax has potential for regeneration of damaged tissues, including during wound healing.

## 1. Introduction

Wounds compromise the integrity of biological tissues, including the skin, mucous membranes and organs. Wound healing is the process by which damaged and dysfunctional cellular structures and tissue layers are replaced [1]. Skin wounds are classified as acute or chronic wounds. Acute wounds proceed through the stages of healing, which are characterized by well-defined signs, within six weeks. Chronic wounds do not undergo normal progression through the stages of healing, and healing is not apparent. The optimal wound-healing process can be divided into four stages: hemostasis, inflammation, proliferation, and maturation [2]. In the hemostasis stage, blood vessels contract immediately after damage, and platelets are activated to form a blood clot. This process stops bleeding and stabilizes the wound area. In the inflammatory phase, white blood cells migrate into the damaged area, and an inflammatory response occurs to remove bacteria and dead cells. Redness, swelling, heat, and pain can accompany this process. Inflammation is an important step in protecting damaged areas and preparing them for healing [3]. Tissue regeneration in the damaged area actively occurs during the proliferative phase. Fibroblasts migrate to the wound site to produce new collagen, and blood vessels are remodeled. In addition, epithelial cells proliferate, and the skin regenerates. In the remodeling and maturation phase, the wound site gradually matures, collagen fibers are rearranged, and strength and elasticity are increased [2]. Over the past decade, researchers have extensively explored innovative therapeutic methods, such as nanotherapeutics, stem cell therapy, and three-dimensional bioprinting. The goal of wound-healing treatment is to increase the accuracy of diagnosis and prognostic prediction to develop individualized treatment strategies. Therefore, the development of cost-effective and noninvasive wound-care therapies is needed to achieve sound wound healing [4].

Polydeoxyribonucleotide (PDRN) is a known wound healing promoter that is extracted from the sperms of *Oncorhynchus mykiss* (salmon trout) or *Oncorhynchus keta* (chum salmon) and contains 50–2000 deoxyribonucleotide base pairs. It is a linear polymer of deoxyribonucleotides in which the monomer units are represented by purine and pyrimidine nucleotides. PDRN was initially shown to effectively promote the regeneration and proliferation of damaged cells, and it has been widely used in cosmetology and dermatology [5]. Previous studies showed that PDRN has properties such as anti-inflammatory activity and the ability to promote angiogenesis and collagen synthesis. Recently, PDRN has been used commercially as an effective compound in cosmetics and drugs and has been approved by the Korea Food and Drug Administration. The use of PDRN in tissue repair and wound healing has also been proposed [6]. In particular, PDRN has been shown to be effective for treating diabetic foot ulcers (DFUs) and chronic wounds in clinical studies [7]. Furthermore, preclinical and clinical studies of PDRN have expanded in various fields, including cartilage and bone regeneration, brain injury treatment and tissue repair [8].

The wound-healing properties of PDRN have been assessed in several studies. PDRN activates the metabolism of fibroblasts and stimulates the production of dermal matrix components [9]. Most studies have demonstrated that the pro-healing effects of PDRN are supported by increased levels of vascular endothelial growth factor (VEGF), a key regulator of angiogenesis [5]. Additionally, PDRN induces the expression of VEGF via an increased binding affinity for adenosine A2A receptors, which stimulates wound healing by promoting angiogenesis. These results indicate that the adenosine A2a receptor plays a role in the mode of action (MoA) of PDRN during wound healing [9]. Moreover, PDRN was shown to increase the expression of proangiogenic factors and inhibit the expression of antiangiogenic factors in an in vitro model of osteoarthritis (OA). PDRN inhibits the pathogenesis of OA by promoting angiogenesis and wound healing, highlighting its potential as a novel regenerative medicine for wound healing in OA, and further studies on this topic are warranted [10].

The placenta is a temporary fetomaternal organ. In the past, this tissue was discarded as medical waste after delivery. However, the placenta has recently been highlighted as an alternative resource in regenerative medicine [11]. In particular, placenta-derived mesenchymal stem cells (PD-MSCs) have many advantages, including their ability to self-renew effectively, their strong immunosuppressive effects and multilineage differentiation potential, and their lack of ethical concerns. Additionally, the immunomodulatory properties of PD-MSCs play crucial roles in the healing of diabetic dermal wounds, making them promising options for treating diabetes-related wounds and DFUs [12]. However, the effect of PDRN derived from PD-MSCs on wound healing is still unclear.

Tonello et al. reported that PDRN extracted from the human placenta has potential for tissue regeneration and wound healing [13]. The DNA fragments in PDRN interact with adenosine receptors, particularly A2A receptors, which play roles in cellular growth, differentiation, and anti-inflammatory processes. Human placenta-derived PDRN has been shown to increase the growth of various cell types and stimulate the synthesis of nucleic acids [14]. Nevertheless, separating PDRN from the placenta at the tissue level is difficult, and many relevant variables differ between individuals.

However, PDRN from human PD-MSCs can be easily extracted and purified from stem cells and is thus expected to have therapeutic efficacy similar to that of PDRN from the placenta. Moreover, PDRN may make substantial therapeutic contributions in regenerative medicine due to its various biological mechanisms. In this study, we assessed the optimal size and dose of PDRN derived from human PD-MSCs via a sonicator system and evaluated its effect on wound healing in in vitro and in vivo models.

## 2. Results

### 2.1. Human PDRN (UNIPlax) Derived from PD-MSCs Affects Cell Migration

We first aimed to determine the experimental conditions for gDNA isolation from human PD-MSCs via a sonication system. Placentas obtained at term (37 gestational weeks) from women who did not experience any obstetric, perinatal, or surgical problems were used. The IRB at CHA General Hospital, Seoul, Korea, approved the sample collection and usage for the study (IRB 07–18). Written informed consent was obtained from all the women who participated. PD-MSCs were isolated via a previously described method [13]. gDNA was isolated from PD-MSCs. The purity and DNA concentration of PDRN derived from PD-MSCs were 2.0 ± 0.02 and 925.6 ± 5.2 µg/mL, respectively. We subsequently performed ultrasonic sonication to fragment the DNA and confirmed through agarose gel electrophoresis that PDRN (UNIPlax) derived from PD-MSCs has a small size range (<500 bp) (Figure 1A). We evaluated the in vitro cytotoxicity and proliferative activity of UNIPlax using an MTT assay. HaCaT cells were treated with UNIPlax at concentrations of 1, 5, 50, and 100 ng/mL, and their proliferative activity was measured after 24 h. We observed no significant cellular toxicity at concentrations up to 100 ng/mL (Figure 1B).

To determine whether UNIPlax promotes wound healing, we examined cell migration in the human keratinocyte line HaCaT. The migration of HaCaT cells treated with UNIPlax (1, 5, 50, or 100 ng/mL) for 24 h was measured (Figure 1C). The 1 ng/mL and 5 ng/mL UNIPlax-treated HaCaT cell groups presented wound closure percentages of 61.1 and 60.8%, respectively, and these results were significantly different from those of the sham group (*p* < 0.05). UNIPlax had an effect at low concentrations and did not have a dose-dependent effect (Figure 1D). These results suggest that UNIPlax has the potential to exert therapeutic effects on wound healing.

### 2.2. UNIPlax Promotes the Angiogenic Ability of HUVECs

Angiogenesis is essential for several cellular processes, including growth, differentiation, death, and tissue repair. PDRN can stimulate angiogenesis and facilitate wound healing [10]. Thus, we investigated the effects of UNIPlax on angiogenesis. Human umbilical vein endothelial cells (HUVECs) were treated with 1, 5, 50, or 100 ng/mL, and recombinant VEGF was used as a positive control. As shown in the representative images, compared with the control HUVECs, the HUVECs treated with 1 ng/mL or 5 ng/mL UNIPlax presented increased tube length and more branches (Figure 2A). After approximately 24 h of VEGF and UNIPlax treatment, we analyzed the total length of the formed tubes and the number of branches, and among the UNIPlax-treated groups, the groups treated with 1 ng/mL and 5 ng/mL UNIPlax presented greater increases than did the groups treated with recombinant VEGF. Similar to the results of the cell migration assay, there was no significant effect at high doses, including 50 ng/mL and 100 ng/mL (Figure 2B,C). These findings indicate that UNIPlax increases the ability of vascular endothelial cells to form blood vessels and to construct branches.

### 2.3. UNIPlax Increases the Expression of VEGF Through the A2A Receptor

PDRN has been reported to induce the expression of VEGF by binding to adenosine A2A receptors [15], thereby promoting angiogenesis and accelerating wound healing. Our results revealed that UNIPlax promotes angiogenesis in HUVECs while increasing wound healing in HaCaT cells. Yang et al. reported that keratinocyte-derived VEGF targets other cell types as well as dermal endothelial cells [16]. Therefore, we examined whether UNIPlax treatment increases VEGF expression via the A2A receptor. As shown in Figure 3, the mRNA levels of VEGF significantly increased in all UNIPlax treatment groups (1, 5, 50, and 100 ng/mL), whereas the A2A receptor mRNA levels decreased in all groups except for the 5 ng/mL treatment group (Figure 3A,B).

To investigate the underlying mechanisms, we examined whether the effects of PDRN are mediated through adenosine A2a receptors by using the adenosine A2a receptor antagonist 3,7-dimethyl-1-propargylxanthine (DMPX). Thus, we analyzed the mRNA levels of VEGF and the A2A receptor in HaCaT cells. The cells were treated with DMPX and/or UNIPlax for 24 h. Single or combination treatment with UNIPlax and DMPX did not change the A2A receptor mRNA levels (Figure 3C). UNIPlax treatment significantly increased VEGF mRNA expression, whereas cotreatment with DMPX and UNIPlax abolished the increase induced by UNIPlax (Figure 3D). These results suggest that UNIPlax can activate a signaling pathway that induces VEGF expression through the A2A receptor.

### 2.4. Efficacy of UNIPlax in an In Vivo Wound Healing Model

To confirm the therapeutic effects of UNIPlax, we used an in vivo wound-healing model. Eight-week-old hairless mice were anesthetized, and after aseptic preparation of the surgical site, 6-mm full-thickness skin wounds were created on the dorsal area, followed by intradermal injections of saline, UNIPlax (5, 50, 100 ng/mL) or Rejuran (50 ng/mL), which is a filler product made from purified polynucleotides, as a representative skin regeneration booster, at four points around the wound border (Figure 4A). Rejuran experiments have shown that polynucleotides promote the growth of human corneal fibroblasts and increase the repair of ultraviolet B (UVB)-damaged dermal fibroblasts [17]. As shown in Figure 4, wound closure was significantly faster in the group treated with UNIPlax than in the saline-treated group (Figure 4B,C). From day 1 after injection, the wound size decreased in the UNIPlax (5, 50, 100 ng/mL) and Rejuran (50 ng/mL) treatment groups, and the wound recovery rate in the UNIPlax (5 ng/mL) treatment group was greater that in the saline-treated group until day 8. The size of the wound area in the UNIPlax (50, 100 ng/mL) and Rejuran (50 ng/mL) treatment groups did not markedly differ from that in the saline-treated group until day 8, but the wound area was notably smaller in the UNIPlax (50 ng/mL) treatment groups (Table 1). The remaining wound areas on day 2 were approximately 73.2%, 65.3%, 70.1%, 75.4%, and 82.9% for the saline, UNIPlax (5, 50, 100 ng/mL)-treated, and Rejuran (50 ng/mL)-treated groups (Figure 4D). In addition, the remaining wound area on day 4 was approximately 55.5%, 39.9%, 58.2%, 50.2%, and 46.4% for the saline, UNIPlax (5, 50, 100 ng/mL), and Rejuran (50 ng/mL) groups, respectively (Figure 4E). Similar to the results of the in vitro experiments, a dose-independent response was observed, with the greatest effect in the UNIPLax 5 ng/mL group. These results suggest that UNIPlax is a potential treatment for promoting wound healing.

### 2.5. UNIPlax Accelerates the Wound-Healing Process

Interestingly, normal histological skin characteristics were observed on the 8th day after wound induction. Multiple epithelial layers were observed on the intact basement membrane in the epidermal layer. Normal tissue was found in the dermal layer, along with hair follicles, an adipose tissue layer, and blood vessels. In contrast, the saline group presented a substantial wound gap, which included the wound margin. This gap was filled with granulation tissue, which had many active myofibroblasts. Additionally, newly formed blood capillaries were observed (Figure 5A). Compared with the saline- and 50 ng/mL Rejuran-treated groups, the UNIPlax 5 ng/mL-treated group presented a significantly decreased granulation tissue area (*p* < 0.001 and *p* < 0.01); however, compared with the sham group, the UNIPlax 50 ng/mL- and 100 ng/mL-treated groups were not significantly different (Figure 5B). Interestingly, granulation tissue, which is a contractile tissue, is characterized histologically by the presence and proliferation of fibroblasts at wound sites (Figure 5C). Although the reactivity of PCNA in granulation tissues in the UNIPlax 5 ng/mL treatment group was lower than that in the other UNIPlax treatment groups, there were no significant differences compared with that in the saline-treated group (Figure 5D). These results indicate that low-dose UNIPlax treatment was most effective for wound healing.

### 2.6. UNIPlax Increases Collagen Deposition During the Wound-Healing Process

The normal wound-healing process relies on a delicate balance between collagen synthesis and collagen breakdown by matrix metalloproteinases (MMPs). During extracellular matrix (ECM) remodeling, the adult wound bed is dominated by the deposition of type I collagen. To determine whether UNIPlax affects collagen fiber synthesis, we performed Masson’s trichrome staining to assess collagen deposition. On day 8, the proportion of collagen was high in normal tissue, and collagen formation was not observed in the granulation tissue area (Figure 6A). Collagen deposition was significantly greater in the UNIPlax 5 ng/mL treatment group than in the saline treatment group. Moreover, compared with the other groups, the UNIPlax 5 ng/mL treatment group presented increased collagen production and a decreased granulation tissue area, whereas the other groups presented blood clots and were still in the early stages of wound healing (Figure 6B). These results suggest that low-dose UNIPlax (5 ng/mL) treatment facilitates the deposition of collagen in skin wounds during wound healing.

### 2.7. UNIPlax Regulates the Expression of VEGF in Skin Wound Tissues In Vivo

As in the in vitro experiments, quantitative real-time PCR was performed to assess A2A receptor and VEGF expression in the tissues treated with UNIPlax (5, 50, or 100 ng/mL) or Rejuran (50 ng/mL). There was no difference in the expression of the A2A receptor across the groups (Figure 7A). In contrast, the VEGF mRNA level was significantly elevated in the wounds of the UNIPlax (5 ng/mL) treatment group (Figure 7B). These results indicate that UNIPlax treatment can modulate A2A receptor activity and upregulate VEGF mRNA expression in tissues in vivo. These results suggest that a low dose of UNIPlax increases the expression of VEGF via the A2A receptor, although this effect does not depend on the dose.

## 3. Discussion

PDRN was first characterized as a tissue-repair stimulating agent extracted from the human placenta. The molecular weight distribution of the PDRN pool within the formulation, as determined through electrophoresis and high-performance liquid chromatography (HPLC), revealed a specific range of 50–2200 base pairs, confirming that PDRN is the active component [13]. Although there are different sources of PDRN, including the human placenta and the sperm of *Oncorhynchus mykiss* (salmon trout) and *Oncorhynchus keta* (chum salmon), previous studies have clearly shown that PDRN from these different sources shares similar properties in terms of promoting wound healing through the PDRN-A2A pathway in skin disorders and other diseases [14,18]. In this study, we demonstrate the development of an alternative source of PDRN, UNIPlax, which is produced by extracting and fragmenting genomic DNA (gDNA) from MSCs derived from the placenta. Although the separation and purification processes for this biomaterial still need to be further developed, this approach could address problems related to material safety and variations in yield across batches related to placenta-derived PDRNs because of the need for a system that meets GMP production requirements for safety.

In most studies, human MSCs have been recognized for their ability to proliferate and differentiate into skin cells, aiding in the restoration of injured or dead cells. Additionally, the ability of MSCs to function via paracrine pathways accelerates cell regeneration as well as wound healing by supporting the skin structure and the activity of the ECM [3]. Moreover, the transplantation of MSCs was reported to promote cutaneous wound healing through paracrine signaling downstream of VEGF, which is released by MSCs [19]. The inflammation and oxidative stress that occur during wound healing not only attract bone marrow-derived MSCs to the wound site, promoting their self-renewal and proliferation, but also support the healing process through the differentiation and stimulation of blood vessel formation [20]. In another study, transplanted PD-MSCs localized to the wound tissue, integrated into the recipient vasculature and increased angiogenesis by secreting proangiogenic molecules, including VEGF, HGF and IGF-1, at bioactive levels [21]. In our previous reports, we transplanted PD-MSCs to promote tissue regeneration and repair in degenerative diseases such as polycystic ovary syndrome (PCOS), metabolic dysfunction-associated steatohepatitis (MASH), and age-related macular degeneration (AMD) [22,23]. For example, PD-MSCs normalized ovarian function in an ovariectomized rat model by increasing HGF secretion through the activation of Wnt signaling [24]. Additionally, PD-MSCs induced the expression of VEGF and VEGFR2 and the activation of their downstream signaling pathways, such as the PI3K/AKT signaling pathway, in injured ovarian tissues [25]. Therefore, PD-MSCs and PDRN (UNIPlax) derived from PD-MSCs are expected to have effects on a wound healing model.

Numerous previous studies have demonstrated that PDRN functions as a growth promoter in the regeneration of diverse tissues, such as wounded skin, DFUs, osteoarthritic tissues, liver tissues after acute liver injury and tissues subjected to ischemia-reperfusion injury [26,27]. Treatment with UNIPlax increased cell migration without inducing cytotoxicity in HaCaT cells and increased tube length and branch numbers in HUVECs, although these effects were not dose dependent. In other studies, PDRN has been shown to exert opposite effects on ERK activity in HDFs and HEK293 cells. HEK cell migration was greatest when the cells were treated with 1 μg/mL PDRN but was inhibited in a dose-dependent manner at concentrations ≥ 50 μg/mL. These differences might have occurred because the optimal PDRN concentrations are different in HDF and HEK cell types [28]. In addition, Other studies have demonstrated that PDRN increased cell migration in HDF cells in a dose-dependent manner, while in epithelial cells, it promoted migration at low concentrations. Therefore, the reactivity of PDRN seems to be different according to cell type. Despite the efficacy of UNIPlax in skin keratinocytes and vascular endothelial cells, increases in tissue regeneration can be achieved not only through wound treatments but also by the use of other cell types. Thus, the function of PDRN have a different range of reactive concentrations for each cell type, and it could be involved at the tissue level via diverse signaling pathways, where various cell types are mixed.

Notably, previous clinical studies have shown that PDRN promotes foot ulcer healing, shortens the time needed for healing, and promotes re-epithelialization in patients with diabetes [29]. Furthermore, PDRN not only increased wound closure quality but also promoted the accumulation of collagen during the wound healing phase [10]. In the present study, the effects of UNIPlax on skin wounds were investigated in vivo, and the possible underlying mechanism was examined by investigating the bioactivities of skin tissues. In the early stage of wound healing including the inflammatory phase, UNIPlax exhibited a faster regenerative process. In the inflammatory phase, which primarily involves the activation of the innate immune system, neutrophils and monocytes rapidly migrate to the injured skin. Similar to PDRN, UNIPlax may have the potential for anti-inflammatory properties, which are manifested through the inhibition of inflammatory cytokines such as TNF-α, IL-6 and IL1β through the activation of adenosine A2a receptor. UNIPlax not only promoted the wound healing process, such as by reducing the granulation tissue area, decreasing the number of proliferating dermal fibroblasts and promoting angiogenesis, but also promoted collagen deposition in tissues in vivo. UNIPlax promoted the final stage of tissue remodeling during wound healing, and the myofibroblasts in granulation tissue differentiated into dermal fibroblasts, which are the cells that generate collagen without cell proliferation. Although quantitative analysis was not performed, we confirmed histologically that fat formation and vascular infiltration were also significantly increased with UNIPlax treatment. Moreover, since collagen formation is a key indicator of efficacy in wound healing models and our results revealed a dramatic change in collagen deposition after UNIPlax treatment, elucidation of the mechanism by which UNIPlax increases collagen deposition is crucial. According to preclinical study, PDRN showed organized development of the granulation tissue by increased expression of fibronectin and laminin in diabetes-impaired wound model [30]. Also, other study has been evaluated PDRN accelerated the collagen production through the activation of FAK and JNK [31]. The mechanism of collagen formation by UNIPlax may have secreted a factor from keratinocytes that promotes the recovery of dermal fibroblasts, or they may have directly controlled the production of collagen by the activation of FAK and JNK in dermal fibroblasts.

Owing to the beneficial effects of PDRN, DNA fragments act through the activation of the A2A receptor, which leads to endothelial cell proliferation, migration and the expression of VEGF. VEGF was found to act as a stimulant in cell lines, including fibroblasts, keratinocytes and preadipocytes [32,33]. This study demonstrated that UNIPlax promoted the wound healing process via the activation of A2A receptors and their downstream targets, including VEGF. These results reveal the underlying mechanism of action of UNIPlax and illustrate its potential as a new promising therapeutic approach for tissue regeneration in dermatological applications. The number of HaCaT cells tended to increase in response to UNIPlax treatment across the concentration groups, with the most pronounced effects observed at the tissue level under low-dose UNIPlax (5 ng/mL) treatment. These findings indicate that UNIPlax may serve as a modulator of VEGF expression during wound healing while also facilitating tissue regeneration and recovery by mediating complex interactions among diverse cell types within the tissue microenvironment. Therefore, investigating how UNIPlax functions in different cell types, such as dermal fibroblasts, endothelial cells and myofibroblasts, is necessary. In addition, since PDRN exerts anti-inflammatory effects through A2A receptors, elucidation of the fundamental mechanism of action of PDRN as an immunosuppressive factor is needed in the future.

In conclusion, this study demonstrated that PDRN (UNIPlax) derived from PD-MSCs exerts several effects on cell migration and angiogenesis by increasing the expression of VEGF in vitro and in vivo. These results reveal a novel PDRN from placenta-derived MSCs and highlight its efficacy in promoting the wound healing process. These data support the use of PDRN as a new promising therapeutic tool for tissue regeneration and repair (Figure 8).

## 4. Materials and Methods

### 4.1. DNA Isolation and Fragmentation

PD-MSCs were cultured in alpha-minimum essential medium (α-MEM; HyClone, Logan, UT, USA) containing 10% fetal bovine serum (FBS; Gibco-BRL, Langley, OK, USA), 1% penicillin/streptomycin (P/S; Gibco-BRL), 25 ng/mL FGF-4 (PeproTech, Cranbury, NJ, USA), and 1 μg/mL heparin (Sigma-Aldrich, St. Louis, MO, USA) at 37 °C in a humidified atmosphere with 5% CO_2_. Genomic DNA was extracted from the pellets of harvested PD-MSCs using a 5 min Tissue Extraction Kit following the manufacturer’s protocol (Scinomics, Daejeon, Republic of Korea). The extracted DNA was diluted with nuclease-free water to achieve a final concentration of 100 µg/mL, and 100 µL aliquots were dispensed into 0.2 mL tubes. These tubes were then placed in an indirect ultrasonic disperser (Qsonica, Newtown, CT, USA) with the lids closed and treated under conditions of 50 to 70% amplitude, 10 to 20 s pulse/rest, 600 to 800 W power, and 10 to 30 Hz frequency for 20 to 40 min in chilled water. After this treatment, the DNA was subjected to electrophoresis on a 1% agarose gel and visualized under UV light, revealing a fragment size between 100 and 500 bp.

### 4.2. Cell Viability Assay

For analysis of UNIPlax cytotoxicity, an MTT assay was performed. HaCaT cells (2 × 10^3^ per well) were seeded in 96-well plates with DMEM/F12 medium (1% penicillin/streptomycin, 10% fetal calf serum) and cultured for 2 days. UNIPlax was added at concentrations of 0, 1, 5, 10, 50, and 100 ng/mL, followed by incubation for 24, 48, and 72 h. After each period, 20 μL of MTT solution (5 mg/mL) was added, and the mixture was incubated for 2 h at 37 °C in a humidified atmosphere with 5% CO_2_. The medium was then replaced with 100 μL of DMSO, the mixture was shaken for 3 min, and the absorbance was measured at 562 nm to calculate cell viability (%).

### 4.3. Cell Migration

For analysis of the wound-healing effect of UNIPlax, an in vitro wound healing assay was conducted. HaCaT cells (2 × 10^5^ per well) were seeded in a 6-well plate and grown to confluence. A scratch was made in the center of each well via a 1 mL pipette tip, and the medium was replaced with fresh medium containing UNIPlax at concentrations of 1, 5, 10, 50, and 100 ng/mL. Images of the wound area were captured immediately after 24 h of incubation at 37 °C in a humidified incubator with 5% CO_2_ to measure wound closure. The wound-healing effect of UNIPlax was then determined by comparing the reduction in wound width across different concentrations.

### 4.4. Tube Formation Assay

For analysis of the angiogenic ability of endothelial cells, HUVECs were seeded at a density of 1 × 10^5^ cells on 24-well plates coated with Matrigel (Corning, Inc., Corning, NY, USA; Cat. No. 354248). Simultaneously, UNIPlax was added at concentrations of 1, 5, 10, 50, and 100 ng. The plates were then incubated at 37 °C in a 5% CO_2_ incubator for 24 h. The length of the formed tubular structures was assessed and quantified using ImageJ 1.54 software.

### 4.5. RNA Isolation and Quantitative Real-Time PCR

Total RNA was extracted from cells or mouse skin tissue using TRIzol LS (Thermo Fisher Scientifics, Waltham, MA, USA) according to the manufacturer’s instructions. cDNA was synthesized using Super Script III reverse transcriptase (Invitrogen, Waltham, MA, USA). qRT–PCR was performed on a CFX Connect™ Real-Time System (Bio-Rad, Hercules, CA, USA) with primers and SYBR Green PCR master mix (Roche, Basel, Switzerland). The sequences of the primers used in this study for the quantification of human A2A receptor, human VEGF and mouse VEGF were as follows: hA2A receptor, CTGGCTGCCCCTACACATC (forward), and TCACAACCGAATTGGTGTGGG (reverse); hVEGF, GCCTTGCCTTGCTGCTCTAC (forward), and ACATCCATGAACTTCACCACTTCG (reverse); and mVEGF, CAAACCTCACCAAAGCCAGC (forward), and CACAGTGAACGCTCCAGGAT (reverse). Gene expression was quantified via the 2^−∆∆CT^ method, and all the data were analyzed in triplicate. Each sample was examined in triplicate, with human or mouse GAPDH as the internal control for normalization.

### 4.6. Mouse Model of Wound Healing and Treatment with UNIPlax

All animal experiments were approved by the Institutional Animal Care and Use Committee (IACUC240108) of the CHA Laboratory Animal Research Center in Korea. Seven-week-old female BALB/c nude mice (Orient Bio, Seongnam, Republic of Korea) were maintained in an air-conditioned pathogen-free animal facility at room temperature (21 °C). The mice were anesthetized with avertin (100 mg/kg). After the surgical site was prepared aseptically, two full-thickness skin wounds were created on the dorsal part via a 6-mm biopsy punch. UNIPlax was injected intradermally at four injection sites on the border of the wound skin. The wounds were photographed every day after injection using a digital camera, and the wound area was measured by tracing the wound margin. The wound area was analyzed by calculating the percentage of the current wound with respect to the initial wound area.

### 4.7. Hematoxylin and Eosin Staining

The mouse skin tissues were subjected to paraffin embedding, fixed with 10% neutral buffered formalin (BBC, Washington, DC, USA), and sectioned serially into 4 µm thick slices. Xylene and ethanol were used to deparaffinize the sectioned tissues in a dry oven at 60 °C. Tissues that had been deparaffinized were washed in the sink. After a 7 min immersion in Harris hematoxylin (Leica Biosystems, Wetzlar, Germany), the slides were counterstained with alcoholic eosin Y solution (Sigma-Aldrich). The stained slides were scanned using an Eclipse Ni system (Nikon, Tokyo, Japan). Images of serial sections were analyzed quantitatively.

### 4.8. Masson’s Trichrome Staining

The paraffin block containing the skin was cut into 4-μm sections, and the tissues were deparaffinized in a 60 °C drying oven with xylene and ethanol. Deparaffinized tissues were then washed with 1× phosphate-buffered saline (PBS) and stained with a Masson’s trichrome staining kit (Abcam, Cambridge, UK) following the manufacturer’s instructions. The stained slides were scanned using an Eclipse Ni system (Nikon). Images of serial sections were analyzed quantitatively.

### 4.9. Immunohistochemistry

The tissues were deparaffinized and washed with 1× PBS. The PBS was removed from the tissues, and the samples were placed in a humidified chamber. The blocking solution (DAKO) was applied to the tissues for 1 h at room temperature, and the anti-PCNA antibody was applied to each tissue sample overnight in a cold, 4 °C room. The tissues were then treated for an additional hour at room temperature. After three rounds of washing with 1× PBS at room temperature for 5 min each, the tissues were exposed to a secondary antibody for an hour at room temperature. The samples were washed three times for five minutes in 1× PBS at room temperature. Afterward, the tissues were incubated with the corresponding secondary antibodies (DAKO) for 1 h at room temperature. The color was developed via a liquid diaminobenzidine substrate chromogen system (DAKO). Hematoxylin was used for counterstaining. Three randomly selected fields were acquired using a 40× objective lens for each tissue with an Eclipse Ni system (Nikon). Images were obtained, and positive cells in the dermis or granulation tissue of the mouse skin were counted.

### 4.10. Statistical Analysis

Multiple group comparisons were performed using the nonparametric Kruskal–Wallis test. Post hoc analysis was conducted with the Conover-Iman method, and significance levels were adjusted via BH correction. Statistical significance was denoted as * *p* < 0.05, ** *p* < 0.01, and *** *p* < 0.001.

## Figures and Tables

**Figure 1 ijms-26-01769-f001:**
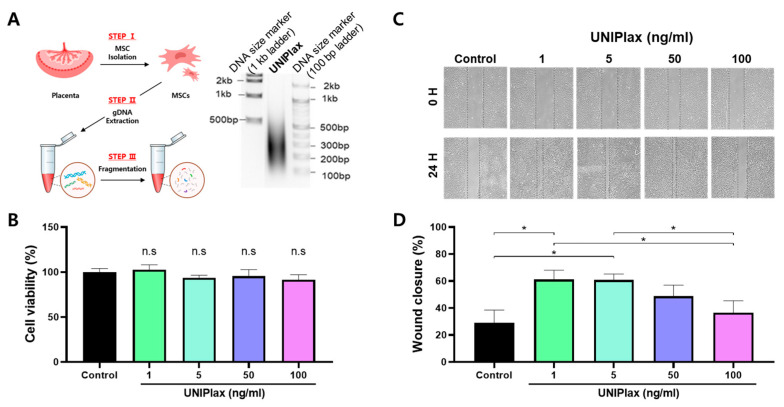
UNIPlax promotes cell migration. (**A**) Schematic diagram of UNIPlax extraction. The DNA size of purified UNIPlax was assessed via gel electrophoresis. (**B**) Cytotoxicity was analyzed in HaCaT cells treated with various doses of UNIPlax. (**C**) Representative images of HaCaT cell migration. The cells were treated with 1–100 ng/mL UNIPlax for 24 h. Scale bars: 25 µm. (**D**) The percentage of wound closure was analyzed via ImageJ 1.54 software. Bars represent the standard error of the mean (SEM). Data from one of two independent experiments are shown (n = 4–6). A nonparametric Kruskal–Wallis test was employed to compare multiple groups, followed by Conover–Iman post hoc analysis with Benjamini–Hochberg (BH) correction for significance testing. * *p* < 0.05, n.s: not significant.

**Figure 2 ijms-26-01769-f002:**
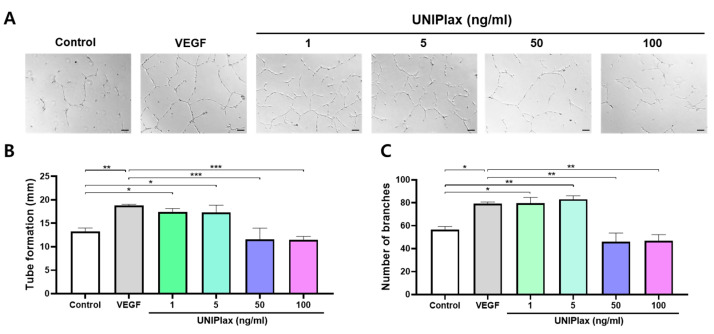
UNIPlax promotes tube formation and branching. (**A**) Representative images of the tube formation assay in HUVECs. The cells were treated with 1–100 ng/mL UNIPlax for 24 h. VEGF (10 ng/mL) was used as a positive control. Scale bars: 100 µm. (**B**,**C**) The total length of newly formed tubes and the number of branches were analyzed via ImageJ software. Bars represent the standard error of the mean (SEM). Data from one of two independent experiments are shown (n = 4–6). A nonparametric Kruskal–Wallis test was employed to compare multiple groups, followed by Conover–Iman post hoc analysis with BH correction for significance testing. * *p* < 0.05, ** *p* < 0.01, *** *p* < 0.001.

**Figure 3 ijms-26-01769-f003:**
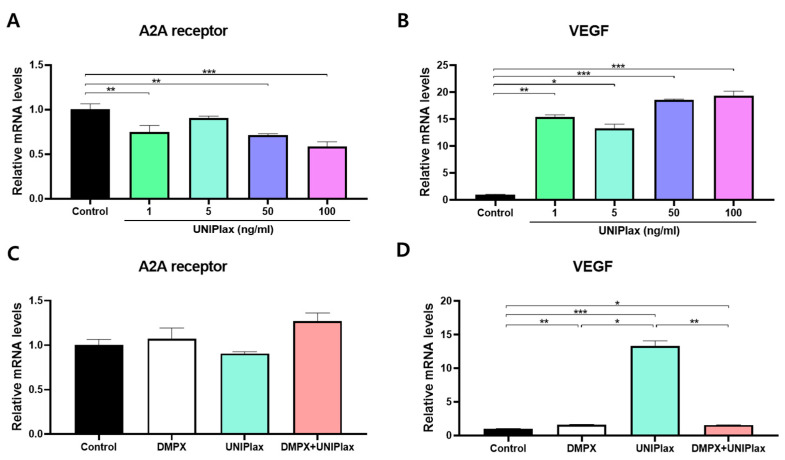
The regulation of VEGF expression by UNIPlax through the A2A receptor. (**A**,**B**) mRNA expression of the A2A receptor and VEGF in HaCaT cells treated with 1–100 ng/mL UNIPlax for 24 h. (**C**,**D**) mRNA expression of the A2A receptor and VEGF in HaCaT cells after treatment with DMPX and/or UNIPlax for 24 h. Bars represent the standard error of the mean (SEM). Data from one of two independent experiments are shown (n = 4–6). A nonparametric Kruskal–Wallis test was employed to compare multiple groups, followed by Conover–Iman post hoc analysis with BH correction for significance testing. * *p* < 0.05, ** *p* < 0.01, *** *p* < 0.001.

**Figure 4 ijms-26-01769-f004:**
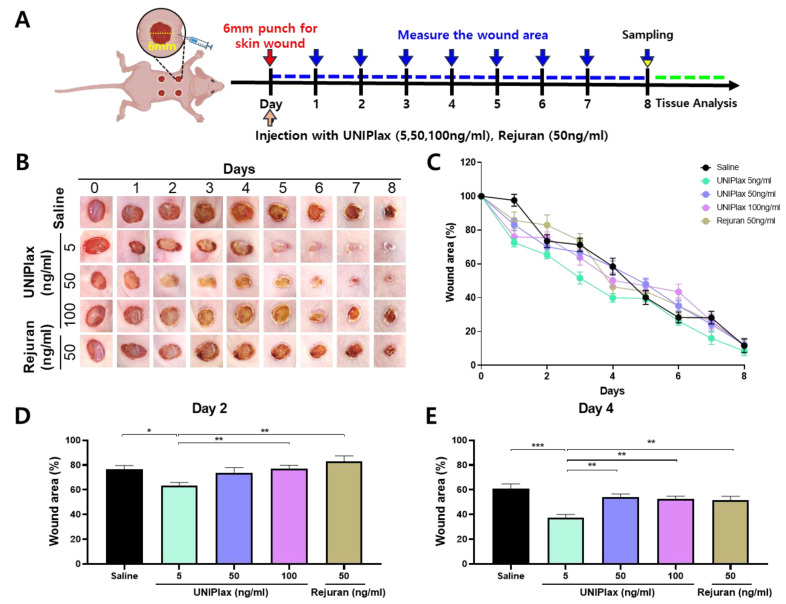
Therapeutic effects of UNIPlax in an in vivo wound-healing model. (**A**) Schematic diagram of the in vivo experiment. Mice with full-thickness skin excision wounds were treated with UNIPlax (5, 50, or 100 ng/mL), Rejuran (50 ng/mL) or saline via intradermal injection from day one post-wounding until sacrifice, as described in the Section 4. (**B**) Representative macroscopic digital images of the wounds treated with UNIPlax (5, 50, or 100 ng/mL), Rejuran (50 ng/mL), or saline are shown at different time points during the wound-closure process. (**C**) The wound area was recorded and is presented as the percentage of the open wound size relative to its original size on the day of wounding. (**D**,**E**) The percentage of the wound area remaining on Days 2 and 4. Bars represent the standard error of the mean (SEM). Data from one of two independent experiments are shown (n = 4–6). A nonparametric Kruskal–Wallis test was employed to compare multiple groups, followed by Conover–Iman post hoc analysis with BH correction for significance testing. * *p* < 0.05, ** *p* < 0.01, *** *p* < 0.001.

**Figure 5 ijms-26-01769-f005:**
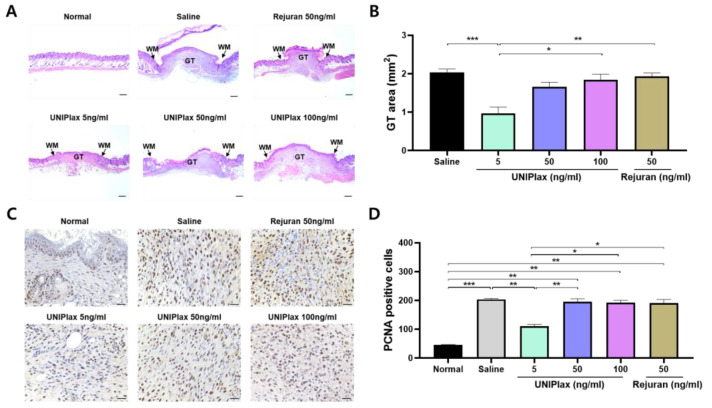
Histological analysis of skin wounds following UNIPlax treatment. (**A**) Representative images of the granulation tissue area in the wound were analyzed after H&E staining. (**B**) The granulation tissue area was measured with a Nikon Eclipse microscope. (**C**) Representative images of tissues immunostained with the anti-PCNA antibody. Scale bar, 10 μm. (**D**) Counts of cells labeled with each antibody in the granulation area from the mice treated with UNIPlax (5, 50, or 100 ng/mL) or Rejuran (50 ng/mL). WM: wound margin, GT: granulation tissue. Bars represent the standard error of the mean (SEM). Data from one of two independent experiments are shown (n = 4–6). A nonparametric Kruskal–Wallis test was employed to compare multiple groups, followed by Conover–Iman post hoc analysis with BH correction for significance testing. * *p* < 0.05, ** *p* < 0.01, *** *p* < 0.001.

**Figure 6 ijms-26-01769-f006:**
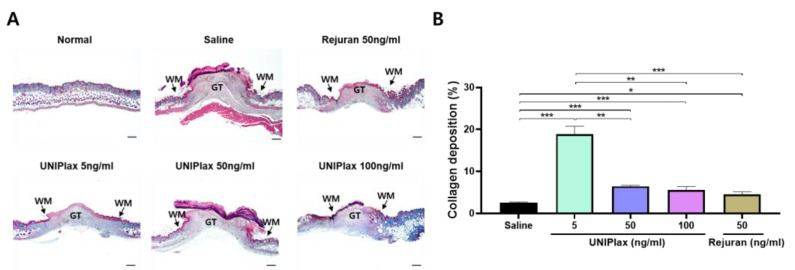
Collagen deposition increased during the wound-healing process after UNIPlax treatment. (**A**) Representative images of collagen deposition in the wound were analyzed after Masson’s trichrome staining. (**B**) The percentage of collagen deposition in granulation tissue was determined via ImageJ software. Scale bars: 500 µm. Bars represent the standard error of the mean (SEM). Data from one of two independent experiments are shown (n = 4–6). A nonparametric Kruskal–Wallis test was employed to compare multiple groups, followed by Conover–Iman post hoc analysis with BH correction for significance testing. * *p* < 0.05, ** *p* < 0.01, *** *p* < 0.001.

**Figure 7 ijms-26-01769-f007:**
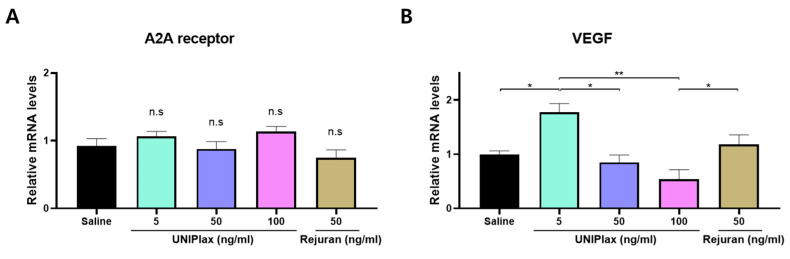
Regulation of VEGF expression by UNIPlax through the A2A receptor. (**A**,**B**) A2A receptor and VEGF mRNA expression in skin wounds treated with UNIPlax (5, 50, or 100 ng/mL) or Rejuran (50 ng/mL). Bars represent the standard error of the mean (SEM). Data from one of two independent experiments are shown (n = 3). A nonparametric Kruskal–Wallis test was employed to compare multiple groups, followed by Conover–Iman post hoc analysis with BH correction for significance testing. * *p* < 0.05, ** *p* < 0.01, n.s: not significant.

**Figure 8 ijms-26-01769-f008:**
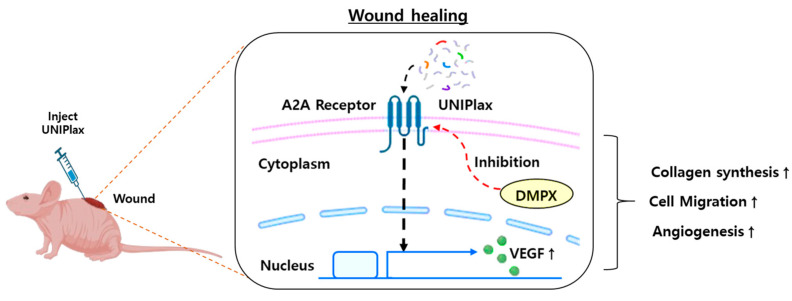
Schematic diagram showing that UNIPlax accelerates the wound-healing process. UNIPlax contains fragmented gDNA from PD-MSCs. UNIPlax promotes collagen synthesis, cell migration, and angiogenesis by increasing VEGF expression through A2A receptor signaling during wound healing.

**Table 1 ijms-26-01769-t001:** Wound area (%) and standard error of the mean wound area by time.

Group	Time (Day)								
	0	1	2	3	4	5	6	7	8
Saline	100	97.6 ± 3.5	73.5 ± 3.9	71.3 ± 4.0	58.5 ± 4.9	40.2 ± 4.1	28.3 ± 3.2	28.2 ± 3.7	11.6 ± 4.0
UNIPlax 5 ng/mL	100	72.5 ± 2.5	65.3 ± 2.3	51.7 ± 3.6	39.9 ± 2.6	39.6 ± 2.1	26.1 ± 2.5	16.0 ± 3.6	8.5 ± 2.7
UNIPlax 50 ng/mL	100	83.2 ± 3.5	70.1 ± 4.2	66.9 ± 3.6	58.2 ± 3.6	48.1 ± 3.4	35.2 ± 2.9	24.3 ± 4.1	12.2 ± 3.9
UNIPlax 100 ng/mL	100	76.0 ± 3.8	75.4 ± 3.4	63.7 ± 4.5	50.2 ± 3.3	47.0 ± 3.7	43.4 ± 4.6	26.2 ± 4.1	12.2 ± 3.5
Rejuran 50 ng/mL	100	85.7 ± 4.9	82.9 ± 6.0	73.7 ± 4.3	46.4 ± 3.9	43.7 ± 3.4	35.2 ± 4.8	25.8 ± 4.5	11.8 ± 3.1

## Data Availability

Data are contained within the article.

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
