# Peer review of "Human Placenta MSC-Derived DNA Fragments Exert Therapeutic Effects in a Skin Wound Model via the A2A Receptor"

_ijms, 2025, doi:10.3390/ijms26041769_

Round 1

Reviewer 1 Report

Comments and Suggestions for Authors

The present manuscript describes the effect of MSC-derived DNA fragments (UNIPlax) on wound healing. The authors describe the methods they use to obtain and characterize the DNA fragments from MSC derived from placenta. They show how UNIPlax does not affect cell viability but increase wound closure in HaCaT cells, additionally they describe that UNIPlax is effective in promoting angiogenesis through the A2A receptor. In an in vivo model, they showed how UNIPlax accelerates wound healing and increases collagen deposition.

The manuscript is well conceived and well conducted. However, there are some issues the authors need to solve before its publication:

1.       It is not clear whether the purification method is purely enough to know that the authors are working only with DNA fragments or the obtained solution contains other bioactive molecules, for instance: proteins and amino acids, among many others.

2.       They attribute the effect of UNIPlax to the DNA fragments, however, it would be helpful that the authors use DNAse-treated UNIPlax to determine its specific activity. They are enourage to discuss this issue. 

3.       It would be helpful for the authors the comparison of their results with those experiments made with a positive control such as a commercially available PDRN. They are invited to discuss this issue.

4.       It is hard to understand how the in vivo and in vitro responses are dose-independent. The authors need to explain in a more detailed manner why UNIPlax at low dose is effective, whole at high dose is not effective.

5.       Also, the authors need to review and correct orthographic mistakes, for instance: “It is co a linear polymer of deoxyribonucleotides and a source of purines and pyrimidines”. Line 60-61.

Author Response

We greatly appreciate the reviewer’s detailed comments on “Human placenta MSC-derived DNA fragments exert therapeutic effects in a skin wound disease model via the A2A receptor”. We revised and added some details as well as reviewer’s comments.

  1. It is not clear whether the purification method is purely enough to know that the authors are working only with DNA fragments or the obtained solution contains other bioactive molecules, for instance: proteins and amino acids, among many others.
  • Author’s response: We thank the reviewer for this criticism. The UNIPlax used in the current method does contain only DNA fragments.

  1. They attribute the effect of UNIPlax to the DNA fragments, however, it would be helpful that the authors use DNAse-treated UNIPlax to determine its specific activity. They are enourage to discuss this issue. 
  • Author’s response: We thank the reviewer once again for reasonable criticism. Similar to the previous suggestion, it could be a helpful point to see the activity by DNase treatment on UNIPlax after increase the purity of the DNA fragment.

  1. It would be helpful for the authors the comparison of their results with those experiments made with a positive control such as a commercially available PDRN. They are invited to discuss this issue.
  • Author’s response: We thank the reviewer once again for reasonable criticism. As suggested by the reviewer, we need experiments to compare using commercially available PDRN as positive control. However, according to recent reports, PDRN promotes cell migration in HDF and has no effect in keratinocytes (SM Shin et al. Mol Med Rep, 2023; D Chae et al, Curr. Issues Mol.Bio, 2025). Therefore, we are planning to find out the cell type-specific efficacy of UNIPlax in the future. We can use PDRN as positive control when in vitro efficacy by cell type such as melanocytes and dermal fibroblasts of UNIPlax.
  • SM Shin, EJ Baek, KH Kim, KJ Kim, EJ Park. Polydeoxyribonucleotide exerts opposing effects on ERK activity in human skin keratinocytes and fibroblasts. Mol Med Rep. 2023 Jun 20;28(2):148.
  • D Chae, SW Oh, YS Choi, DJ Kang, CW Park, J Lee and WS Seo. First Report on Microbial-Derived Polydeoxyribonucleotide: A Sustainable and Enhanced Alternative to Salmon-Based Polydeoxyribonucleotide. Curr. Issues Mol. Biol. 2025, 47, 41.

  1. It is hard to understand how the in vivo and in vitro responses are dose-independent. The authors need to explain in a more detailed manner why UNIPlax at low dose is effective, whole at high dose is not effective.
  • Author’s response: We greatly appreciate the reviewer bringing up this important point. In previous studies, the effect of PDRN on HDF proliferation at the high concentration did not seem to be remarkable. However, the low concentration of PDRN improved cell proliferation (KH Hwang et al, Mol Med Rep, 2018). In another studies, PDRN increased cell migration in a dose-dependent manner in HDF cells, but in epithelial cells, it promoted cell migration at low concentrations (Y Koo et al, Materials Science and Engineering C, 2016). Although we do not know exactly yet, the function of PDRN have a different range of reactive concentrations for each cell type, and it could be involved at the tissue level via diverse signaling pathways, where various cell types are mixed. This contents added into the revised manuscript (line336-338, line341-343).
  • KH Hwang, JH Kim, EY Park, SK Cha. An effective range of polydeoxyribonucleotides is critical for wound healing quality. Mol Med Rep. 2018 Dec;18(6):5166-5172.
  • Y Koo, Y Yun. Effects of polydeoxyribonucleotides (PDRN) on wound healing: Electric cell-substrate impedance sensing (ECIS). Materials Science and Engineering C 69 (2016) 554–560.

  1. Also, the authors need to review and correct orthographic mistakes, for instance: “It is co a linear polymer of deoxyribonucleotides and a source of purines and pyrimidines”. Line 60-61.
  • Author’s response: We apologize for the defects in our original manuscript. There was a mistake in writing the content. We have thoroughly revised the manuscript to " It is a linear polymer of deoxyribonucleotides in which the monomer units are represented by purine and pyrimidine nucleotides” in the text ((line 61-62)).

Reviewer 2 Report

Comments and Suggestions for Authors

The manuscript by Lee et al. focuses on the extraction of placenta-derived PDRN and determination of its effects in in vitro and in vivo model of wound healing. The aim of the study is important and interesting, however, the presented results seem to be rather preliminary and do not answer the question about the mode of action of obtained PDRN in the cells and tissues.

Major remarks:

1/ The authors claim that an increase in VEGF has been shown as a result of adenosine A2A receptor stimulation by PDRN, but from the results of their study one can conclude that in the presence of PDRN, the expression of A2A receptor is decreased (in vitro) or slightly and non-significantly increased (in vivo). How would you explain that? Did you check ecto-adenosine concentration in the extracellular environment? Is there enough adenosine to activate the receptors? Is there any activity of ecto-5’-nucleotidase to produce adenosine? In my opinion, the involvement of A2A receptor in the presented wound healing models is doubtful, and the conclusion that it is the mode of action is not supported by the results. Moreover, how was it confirmed that PDRN interacts with A2AR with high affinity?

2/ The authors do not discuss why only the low concentration (1 and 5ng/mL) of PDRN exerts beneficial effects on wound healing process, despite the fact that all the concentrations tested were not toxic and did not decrease the cells viability.

3/ In conclusions, line 385 and further, one can read “this study demonstrated that PDRN (UNIPlax) derived from PD-MSCs exerts several effects on cell migration and angiogenesis by increasing the number of VEGF A2A receptors in vitro and in vivo” which is completely misleading. What are “VEGF A2A receptors”? Again, the conclusion that the mode of action of wound healing is through A2AR activation is not supported by the results.

Minor remarks:

1/ In the introduction, line 101 and further, the authors state that “separating PDRN from the placenta at the tissue level is difficult, and many relevant variables differ between individuals. However, PDRN from human PD-MSCs can be easily extracted and purified from stem cells”. What about the variables of such isolation? MSCs are commonly accepted as a very heterogenous population of cells, so the PDRN extracted from them will be also variable.

Comments on the Quality of English Language

The English revision is recommended.

Author Response

We greatly appreciate the reviewer’s detailed comments on “Human placenta MSC-derived DNA fragments exert therapeutic effects in a skin wound disease model via the A2A receptor”. We revised and added some details as well as reviewer’s comments.

  1. The authors claim that an increase in VEGF has been shown as a result of adenosine A2Areceptor stimulation by PDRN, but from the results of their study one can conclude that in the presence of PDRN, the expression of A2A receptor is decreased (in vitro) or slightly and non-significantly increased (in vivo). How would you explain that? Did you check ecto-adenosine concentration in the extracellular environment? Is there enough adenosine to activate the receptors? Is there any activity of ecto-5’-nucleotidase to produce adenosine? In my opinion, the involvement of A2A receptor in the presented wound healing models is doubtful, and the conclusion that it is the mode of action is not supported by the results. Moreover, how was it confirmed that PDRN interacts with A2AR with high affinity?
  • Author’s response: We appreciate these insightful comments. Although we did not directly analyze whether PDRN binds to A2AR or whether adenosine has quantitative analysis, but we analyzed the mRNA levels of VEGF and the A2A receptor in HaCaT cells were treated with DMPX and/or UNIPlax. 3,7-Dimethyl-1-propargylxanthine (DMPX) has been used in the MoA (mode of action, MoA) PDRN studies as a caffeine analog that acts as an antagonist to A2 adenosine receptors. As shown Figure 3D, the effects of UNIPlax were abolished by the concomitant incubation with DMPX. In a represent example study by Thellung et al. (1999), PDRN and adenosine were compared in primary human skin fibroblasts, and both induced cell growth. However, the effects of PDRN were blocked when incubated with the adenosine A2A receptor antagonist, 3,7-Dimethyl-1-propargylxanthine (DMPX), which has a stronger affinity for the A2A receptor than for the A2B receptor. This suggests that PDRN can acts on the A2A receptor, potentially serving as a pro-drug that generates active deoxyribonucleotides, nucleosides, and bases, which interact with the A2A receptor to produce pharmacological effects (Squadrito F et al, Front Pharmacol, 2017; Irrera N et al, Front Pharmacol, 2018).
  • Thellung, S., Florio, T., Maragliano, A., Cattarini, G., and Schettini, G. (1999). Polydeoxyribonucleotides enhance the proliferation of human skin fibroblasts: involvement of A2 purinergic receptor subtypes. Life Sci. 64, 1661–1674.
  • Squadrito F, Bitto A, Irrera N, Pizzino G, Pallio G, Minutoli L, Altavilla D. Pharmacological Activity and Clinical Use of PDRN. Front Pharmacol. 2017 Apr 26;8:224.
  • Irrera N, Arcoraci V, Mannino F, et al. Activation of A2A receptor by PDRN reduces neuronal damage and stimulates WNT/β-CATENIN driven neurogenesis in spinal cord injury. Front Pharmacol. 2018;9:506.

  1. The authors do not discuss why only the low concentration (1 and 5ng/mL) of PDRN exerts beneficial effects on wound healing process, despite the fact that all the concentrations tested were not toxic and did not decrease the cells viability.
  • Author’s response: We thank the reviewer for this criticism. In previous studies, the effect of PDRN on HDF proliferation at high concentrations did not show significant results. However, at lower concentrations, PDRN enhanced cell proliferation (KH Hwang et al, Mol Med Rep, 2018). Other studies have demonstrated that PDRN increased cell migration in HDF cells in a dose-dependent manner, while in epithelial cells, it promoted migration at low concentrations (Y Koo et al, Materials Science and Engineering C, 2016). Although the exact mechanisms are still unclear, it is believed that PDRN may have different effective concentrations for various cell type, with diverse signaling pathways involved at the tissue level, where multiple cell types communication. This contents added into the revised manuscript (line336-338, line341-343).
  • KH Hwang, JH Kim, EY Park, SK Cha. An effective range of polydeoxyribonucleotides is critical for wound healing quality. Mol Med Rep. 2018 Dec;18(6):5166-5172.
  • Y Koo, Y Yun. Effects of polydeoxyribonucleotides (PDRN) on wound healing: Electric cell-substrate impedance sensing (ECIS). Materials Science and Engineering C 69 (2016) 554–560.

  1. In conclusions, line 385 and further, one can read “this study demonstrated that PDRN (UNIPlax) derived from PD-MSCs exerts several effects on cell migration and angiogenesis by increasing the number of VEGF A2A receptors in vitro and in vivo” which is completely misleading. What are “VEGF A2A receptors”? Again,the conclusion that the mode of action of wound healing is through A2AR activation is not supported by the results.
  • Author’s response: We apologize for the defects in our original manuscript. There was a spelling mistake in writing the content. We have thoroughly revised the manuscript to " this study demonstrated that PDRN (UNIPlax) derived from PD-MSCs exerts several effects on cell migration and angiogenesis by increasing the expression of VEGF in vitro and in vivo” in the text (line 380-382).

Minor remarks:

  1. In the introduction, line 101 and further, the authors state that “separating PDRN from the placenta at the tissue level is difficult, and many relevant variables differ between individuals. However, PDRN from human PD-MSCs can be easily extracted and purified from stem cells”. What about the variables of such isolation? MSCs are commonly accepted as a very heterogenous population of cells, so the PDRN extracted from them will be also variable.
  • Author’s response: We thank the reviewer once again for reasonable criticism. We have developed a method for high purity separation of placental chorionic plate-derived mesenchymal stem cells (PD-MSCs) and are under development as stem cell therapy for polycystic ovarian syndrome and aged related macular degeneration. Especially,  we established a PD-MSCs cell banking system at the GMP facility based on standard of protocol (SOP) for isolation and cultivation of PD-MSCs from normal term placenta. In addition, DNA fragments were consistently obtained from cells isolated and cultured in the GMP facility. 

Round 2

Reviewer 2 Report

Comments and Suggestions for Authors

The authors addressed all my comments and remarks, the corrections that were made improved the overall quality of the manuscript.

Author Response

The authors addressed all my comments and remarks, the corrections that were made improved the overall quality of the manuscript.

Author’s response: We thanks for the comments in our manuscript. We have thoroughly revised the manuscript to improve the quality of the data. In addition, there have been many changes throughout the manuscript to improve the quality, accuracy and readability of the writing. We hope the reviewer willingly evaluate the revised manuscript.   
